# Molecular Characterization of *Chlamydia trachomatis* Infection and Its Impact on Sperm Characteristics of Moroccan Infertile Men

**DOI:** 10.3390/idr17060135

**Published:** 2025-10-22

**Authors:** Mariame Kabbour, Modou Mamoune Mbaye, Bouchra Ghazi, Achraf Zakaria, Rajaa Ait Mhand, Noureddine Louanjli, Moncef Benkhalifa

**Affiliations:** 1Laboratory of Chemistry-Physics and Biotechnology of Biomolecules and Materials (LCP2BM), Faculty of Science and Technology Mohammedia, Hassan II University of Casablanca, Casablanca 28806, Morocco; rajaamhand@yahoo.fr; 2Laboratory of Medical Analyses, Reproductive Biology, Labomac, Casablanca 20100, Morocco; mbayeass87@gmail.com (M.M.M.); azakaria@um6ss.ma (A.Z.); 3Immunopathology-Immunotherapy-Immunomonitoring Laboratory, Faculty of Medicine, Mohammed VI University of Health and Sciences (UM6SS), Casablanca 82403, Morocco; bghazi@um6ss.ma; 4Mohammed VI Center for Research & Innovation (CM6RI), Rabat 10112, Morocco; 5Research Laboratory of Microbiology, Infectious Diseases, Allergology and Pathogen Surveillance (LARMIAS), Faculty of Medicine, Mohammed VI University of Health and Sciences (UM6SS), Casablanca 82403, Morocco; 6Laboratory of Medical Analyses, Reproductive Biology, Labomac, African Fertility Clinic AFC, Irifiv Center, Casablanca 20100, Morocco; n.louanjli@gmail.com; 7Reproductive Medicine, Reproductive Biology and Genetics, University Hospital and School of Medicine and PERITOX Laboratory, Picardie University Jules Verne, 80025 Amiens, France; benkhalifamoncef78@gmail.com

**Keywords:** *Chlamydia trachomatis*, male infertility, qPCR

## Abstract

**Background/Objectives**: Infections of the urogenital tract have experienced renewed interest in recent years, due to their frequency and also their impact on sperm parameters and the fertilizing quality of spermatozoa. *Chlamydia trachomatis* (CT) represents an intracellular microorganism responsible for sexually transmitted infections (STIs) in men and women. A reliable method of diagnosing this infection is therefore necessary because of the rapid onset of infection and the increase in STI-related diseases and their treatment costs. **Methods**: We analyzed 2371 semen samples from infertile men and detected the presence anti-CT IgG antibodies by Enzyme-Linked Immunosorbent Assay (ELISA), followed by real-time PCR confirmation of CT DNA whose target is the lipopolysaccharide (LPS). We assessed the effect of CT infections on characteristic parameters of sperm quality, including concentration, motility, viability, and morphology. The impact on sperm DNA quality was assessed by DNA fragmentation index (S) and decondensation of chromatin index (SDI) by the TUNEL technique. **Results**: Analysis of the results showed significant differences in mobility, concentration, and morphology (*p* < 0.05) between the control group, positive CT infection with normal spermiogram status (CT+/Normal SG) group, and positive CT infection with abnormal spermiogram status (CT+/Abnormal SG) group. A significant increase in the DFI and the SDI was found between the control group and the case groups, respectively (*p* < 0.01). **Conclusions**: Our results confirm that CT infection is associated with significant alterations in sperm parameters and sperm DNA quality. Regular CT screening by qPCR should be encouraged in couples suffering from unexplained infertility.

## 1. Introduction

*Chlamydia trachomatis* (CT) is an obligate intracellular bacterium responsible for chlamydial infection, one of the most common sexually transmitted infections (STIs). It is the most reported infection worldwide with 127 million cases annually [1]. Genital CT infection remains a common cause of pathology in both men and women causing urethritis, epididymitis, prostatitis, cervicitis, pelvic inflammation, and tubal infertility [2].

CT is an intracellular bacterium with a particular development cycle in which it takes two forms: the inactive elementary body and the infectious reticulated body, enabling continuous sexual transmission between the two partners [3].

CT infection has been widely associated with infertility in women, but much less is known about its impact on male fertility; this causal link in men remains controversial. The prevalence of CT in infertile men remains variable in the literature (from 1% to 30%) [4,5]. A substantial proportion of men suffering from bacterial infection-induced infertility are asymptomatic, with reported prevalence rates of asymptomatic infection ranging from approximately 50% in various epidemiological studies [6,7,8]. Although comprehensive nationwide data remain limited, available studies indicate that CT prevalence in Morocco ranges between 3% and 10% among women of reproductive age and between 2% and 8% among sexually active men, depending on the population studied and the detection method used [9].

The development of genital infections and inflammation due STIs can lead to defective spermatogenesis, production of antibodies that attack sperm, or blockages in the seminal tract, all of which can have a negative impact on sperm quality [10]. Numerous studies have revealed that STIs can lead to inflammatory responses and oxidative stress, both of which have been demonstrated to play critical roles in male infertility [11]. Oxidative stress can be described as a lack of balance between the production of reactive oxygen species (ROS) and the antioxidant capacity of the seminal plasma. ROS, when present in excess, can cause damage to cell components like lipids, proteins, and nucleic acids [12]. In this context, sperm DNA fragmentation (SDF) has been one of the significant indicators of oxidative damage and imperfect sperm function. 

Leukospermia—defined by the WHO (sixth edition laboratory manual) as a leukocyte concentration of >10^6^ cells/mL of semen—is one such cause of oxidative damage. It is observed in approximately 10 to 20% of infertile men [13,14] and typically suggests underlying infection or inflammation.

Previous studies have shown that CT infection induces phosphatidylserine externalization and sperm DNA fragmentation [15], and also caused premature sperm death via the lipopolysaccharide [16], and others have shown that this bacterium adheres to spermatozoids during their development [17]. An in vitro study in mice showed that testicular cell lines (Sertoli, Leydig, and germinal) infected with CT showed significant changes in cellular pathways (cell signaling) as well as DNA hypomethylation of spermatozoa [18]. The mechanism of DNA fragmentation has been defined in different studies to involve excessive production of reactive oxygen species (ROS) in leukocytes, induced by the presence of CT lipopolysaccharide [19].

In another study, it was shown that the major outer membrane protein (MOMP) was detected in testicular biopsies of infertile men at a rate of 45.3%; this was confirmed by the presence of a replication marker TC0500 and DNA of CT [20]. Several authors have also shown that this infection seems to be widespread among infertile couples, and it can have a negative effect on the quality of sperm characteristics (sperm count, motility, and morphology), which leads to infertility [21].

Diagnosis of CT infection is based on several approaches and techniques that include the detection of anti-CT antibodies (which do not distinguish between a recent and a previous infection), cell culture (rarely used in the clinic), immunofluorescence detection of chlamydia in several samples (urine, semen, or urethral swab), and amplification of chlamydial DNA by PCR [22,23]. As CT infections are often asymptomatic and have been reported to be a cause of male infertility, we hypothesized that molecular diagnosis (by PCR) would provide a more accurate assessment of the prevalence of infection among infertile males than conventional diagnostic tests. Moreover, as recent findings had suggested that such infections affect sperm quality regardless of conventional semen parameters, we suspected that CT can result in disruption of the integrity of sperm DNA, even in healthy men with normal semen parameters. Our approach therefore aims to achieve a better understanding of the potential role of CT in male reproductive function by combining serological and molecular detection and routine semen analysis with sperm DNA fragmentation assay.

The objective of our study is to evaluate the effect of CT infections on the characteristic parameters of sperm quality and their impact on sperm DNA quality through the evaluation of DNA fragmentation and the decondensation of spermatozoa chromatin after detection of the CT infection by ELISA IgG Anti-CT and its confirmation by the molecular tool.

## 2. Materials and Methods

The study was conducted at the Laboratory of Medical Analysis, Biology of Reproduction, LABOMAC, Casablanca, Morocco. It was conducted between February 2022 and April 2024. A total of 2731 semen samples were collected from subfertile couples, with male partners aged between 24 and 56 years. The samples were classified according to two independent criteria: spermiogram status (normal or abnormal) and CT infection status (positive or negative). This classification resulted in four distinct subgroups: (1) normal spermiogram and CT-negative (control group), comprising 678 patients aged 24–56 years old; (2) normal spermiogram and CT-positive, including 59 patients aged 26–40 years old; (3) abnormal spermiogram and CT-negative, with 346 patients aged 30–50 years old; and (4) abnormal spermiogram and CT-positive, consisting of 273 patients aged 33–46 years old. The abnormal spermiogram groups were characterized by one or more of the following alterations: elevated leukocyte count (>1 million/mL), low sperm concentration (≤15 million/mL), abnormal morphology (<4%), reduced total motility (≤42%), or low sperm viability (≤58%). The abnormal spermiogram/CT-negative group was identified but not included in the detailed analysis, as our primary focus was to investigate the impact of CT infection on sperm quality.

On the other hand, patients with a history of antibiotic consumption in the previous two months; those positive for HIV, HBS Ag, HCV Ab, varicocele, or azoospermia; and those with a history of epidymitis, epididymo-orchitis, hypothyroidism, or hyperthyroidism were excluded from this study.

### 2.1. Sperm Collection and Analysis

The spermiogram–spermocytogram is a first-line examination in the quantitative and qualitative evaluation of semen. Spermiogram refers to all tests performed in the fresh state to evaluate the macroscopic (appearance, volume, liquefaction time, viscosity, and pH) and microscopic (sperm count, motility, and viability) characteristics of the semen, while spermocytogram is the cytological analysis performed on a colored sperm smear to evaluate the morphology of the spermatozoa [24].

The samples were collected, by masturbation, in sterile vials and labeled after 3 to 4 days of sexual abstinence. For the liquefaction step, the samples were stored at 37 °C for between 30 and 60 min. We checked at time intervals of 10 min until liquefaction was complete. The analysis of the characteristic parameters of sperm quality (in the case and control groups) including volume, concentration, motility, viability, and morphology was carried out in accordance with the standards and guidelines of the World Health Organization (WHO) using a sperm analyzer (computer-assisted sperm analysis (CASA), Microptic, Barcelona, Spain) [23]. Sperm viability was assessed using eosin 2% staining. A semen sample was mixed with eosin, and at least 200 spermatozoa were examined under light microscopy. Viable sperm remained colorless, while non-viable sperm stained red. Sperm viability was expressed as the percentage of unstained (viable) spermatozoa. For morphology, at least 2 × 200 spermatozoa were analyzed.

Then, 100 µL of raw semen samples (273 CT-positive with abnormal SG cases and 59 controls) were frozen in liquid nitrogen for further analysis of sperm chromatin structure (SCSA). The remainder was centrifuged, and the seminal plasma was removed and transferred to sterile vials and stored at −20 °C for serological testing. The pellet was resuspended in phosphate-buffered saline (PBS, Sigma-Aldrich, Gillingham, UK) and stored at −70 °C for molecular testing.

### 2.2. Detection of CT Antibodies in Seminal Plasma

Anti-CT ELISA (IgG) (EUROIMMUN, PerkinElmer company, Waltham, Germany) employs antigens from CT. Three different antigen categories were used: an electrophoretically separated antigen extract from CT, chlamydia-cross-reacting LPS (contained in the antigen extract), and highly specific recombinantly produced MOMP antigens. This combination ensured high sensitivity and specificity. Seminal plasma was tested manually according to the manufacturer’s instructions. It was diluted 1/10 in the diluent, then incubated with the CT antigens coated on a 96-well plate between +18 °C and +25 °C for 30 min. After 3 successive washes, 100 µL of the conjugate was added to the wells and incubated for 30 min between +18 °C and +25°. After a new washing, 100 µL of the substrate was added. The reaction was stopped by the stop solution and the optical density was read at 450 nm and a reference wavelength between 620 nm and 650 nm within 30 min of adding the stop solution. CT serology is considered positive if the ratio of the extinction of the control or patient sample over the extinction of the calibrator 2 ≥ 1.1.

### 2.3. Molecular Tests

Detection of CT by real-time PCR using the CT Real-TM PCR kit (Sacace Bientechnologies, Como, Italy) is based on the amplification of the specific region of the pathogen genome using specific primers. The Real-TM *Chlamydia trachomatis* PCR Kit is a qualitative assay that contains the internal control (IC), which should be used in the extraction procedure to control the extraction process of each individual sample and to identify possible reaction inhibition. The Real-TM CT PCR kit uses a “hot start”, which greatly reduces the frequency of non-specific priming reactions. “Hot start” is ensured by a chemically modified polymerase (TaqF), which is activated by heating to 95 °C for 15 min.

DNA extraction from sperm sample was performed using the SaMag STD DNA Extraction Kit (Sacace REF SM007). Total reaction volume was 25 µL, sample volume was 10 µL. The PCR-mix-2-FRT tube was thawed and the tubes with PCR-mix-1-FRT, PCR-mix-2-FRT, and Taq F polymerase were vortexed and then centrifuged briefly; then, we transferred 15 µL of the prepared mixture into each tube. Using barrier spray tips, 10 µL of DNA buffer was added to the tube labeled NCA (Negative Control of Amplification), 10 µL of positive control (Pos C+) was added to the tube labeled C+ (Positive Amplification Control), and 10 µL of an extracted sample of the negative control (C−) was added to the tube labeled C− (Negative Extraction Control). The samples were placed in a programmable thermal cycler (SaCycler–96; Sacace Biotechnologies S.r.l., Como, Italy). The amplifications were carried out according to the following program: an initiation denaturation step at 95 °C for 15 min, followed by 5 cycles comprising: denaturation at 95 °C for 5 s; hybridization at 60 °C for 20 s and elongation at 72 °C for 15 s; and finally, 40 cycles at 95 °C for 5 s, 60 °C for 30 s, and elongation at 72 °C for 15 s. CT was detected on the FAM channel (Green), the internal control DNA on the JOE channel (Yellow)/HEX/Cy3.

### 2.4. Sperm DNA Fragmentation Index Assessment by the TUNEL Technique (DFI)

Sperm DNA integrity was assessed by the TUNEL test using a commercial kit (Roche Diagnostic, Lewes, UK) according to the manufacturer’s recommendations. The principle of the TUNEL technique is to use an enzyme, terminal deoxynucleotransferase (TdT), capable of adding nucleotides to the 3′-OH ends of free DNA. The semen sample was washed twice in phosphate-buffered saline (PBS, Sigma-Aldrich, Gillingham, UK) and adjusted to a concentration of 2 × 10^7^ cells/mL in PBS. The cell suspension was then fixed in PBS containing 2% formaldehyde (Sigma-Aldrich, Gillingham, UK) for 60 min at room temperature. After a double wash with PBS, the sample was centrifuged at 1200 rpm. This step was repeated twice. The prepared slides were immediately analyzed using a fluorescence microscope (Nikon Eclipse 80i, Nikon Corporation, Tokyo, Japan) equipped with appropriate filters. Images were captured using a CCD camera and XytoGen software (Excilone, version 3.8.46, Elancourt, France) [25].

### 2.5. Sperm DNA Decondensation Index Analysis with Aniline Blue (SDI)

Slides of prepared sperm samples were rinsed twice with distilled water and then stained with a 5% aniline blue bath at pH 3.5 for five minutes. They were then quickly rinsed with distilled water, then dehydrated in alcohol baths (70, 90, and 100 °C, one minute each). Reading was done under white light at ×1000 magnification and by immersion. A total of 500 sperm were counted on the slides made from the WHO sample (examined according to WHO spermiogram and spermocytogram guidelines).

### 2.6. Statistical Analysis

Data are presented as mean and 95% confidence interval. Statistical analysis was performed using DataTab software (DATAtab Team, version 2025; Graz, Austria). Quantitative results were analyzed using one-way ANOVA. Levene’s test (F-test) was used to check the homogeneity of variances and the normality of the data as ANOVA assumptions. The two-sided *p* value < 0.05 was determined to be statistically significant. All histograms shown in this article were created with DataTab software.

## 3. Results

In this study, the average age of the selected subfertile men was 34.2 ± 6.4 years with extremes ranging from 24 to 56 years. Apart from the low semen volume observed in patients with CT infection, all volumes of patients in the control group were normal. There was no significant difference in concentration, motility, viability, or morphology between the control group and SG normal/CT-positive group (*p* > 0.05) (Table 1). On the other hand, significant differences were observed in the concentration, mobility, viability, and morphology of spermatozoa between the control and SG abnormal/CT-positive group as well as between SG abnormal/CT-positive and the SG normal/CT-positive groups (*p* < 0.01)) (Table 1). No statistically significant difference was found regarding the concentration of leukocytes between the three groups (Table 1).

The prevalence of CT infection varied depending on the detection method and semen profile category. Using ELISA, CT infection was detected in 159 patients (6.7%), including 102 (4.29%) with normal spermiogram (SG) and 57 (2.41%) with abnormal SG (Table 2). However, ELISA produced one false-positive result. In contrast, molecular detection via real-time PCR revealed CT infection in 173 patients (7.3%) who had tested negative for anti-CT antibodies by ELISA. Among them, 96 (4.05%) had normal SG and 77 (3.25%) had abnormal SG. Notably, all patients who tested positive by ELISA were confirmed positive by real-time PCR, except for one false-positive sample. All PCR analyses were performed in duplicate to ensure result reliability.. In cases of discrepancy or borderline results, the analysis was repeated, and consensus was reached before reporting the final result.These findings highlight the higher sensitivity of real-time PCR and suggest that reliance on serological testing alone may underestimate the true prevalence of CT infection.

In the 332 CT-positive patients, there was a significant increase in DFI and SDI of cases (*n* = 273 patients with abnormal spermiogram) compared to controls (*n* = 59 patients with normal spermiogram) (*p* < 0.01) (Figure 1).

## 4. Discussion

CT infection is very common in both men and women. It is very often asymptomatic; hence the possibility that the number of infected people is twice as large as the number of reported cases [26]. In our study, the average age of men was 34 ± 6.4 years old with extremes ranging from 24 to 56 years old. In the present study, no statistically significant differences were observed between the control group and the SG normal/CT-positive group in terms of sperm concentration (82.47 and 74.28 million/mL), total motility (71.38% and 59.82%), viability (84.59% and 73.65%), and normal morphology (9.27% and 8.42%), respectively (*p* > 0.05) (Table 1). The reason why the actual number of people diagnosed was much lower than the number infected is that CT infection is asymptomatic (68% of cases), which makes diagnosis complicated. In symptomatic patients, CT infection leads to subacute urethritis, characterized by a mild, serous urethral discharge that may occur spontaneously or upon pressure on the urethral canal. The discharge is typically white, gray, or clear, and may only be noticeable after penile “stripping” or in the morning [27].

Significant differences in sperm motility (71.38% and 38.54%), viability (84.59% and 45.93%), concentration (82.47 and 36 million/mL), and normal morphology (9.27% and 1.75%) were observed between the SG normal/CT-positive group and the SG abnormal/CT-positive group, respectively (*p* < 0.01) (Table 1). These results are in agreement with those of Idahl et al., who showed that the decrease in motility, the number of spermatozoa, and the increase in the index of teratozoospermia were significantly associated with the presence of IgM and IgG antibodies against CT in the serum of infertile patients [28]. However, these results differ from those of Gdoura et al., who reported that sperm volume, concentration, count, motility, and morphology were not associated with the detection of DNA CT in semen samples from infertile men [29]. However, our results on the non-significant difference in leucospermia between the control, SG abnormal/CT-positive, and SG abnormal/CT-positive groups remain identical to those of Gdoura who also found non-significant differences [30].

In vitro experiments have demonstrated that *Chlamydia trachomatis* can infect and replicate within primary human Sertoli cells, leading to significant alterations of the cellular cytoskeleton, including disruption of key structural components such as F-actin fibers, vimentin-based intermediate filaments, and microtubules [16,31]. Other findings indicate that CT lipopolysaccharide (LPS) may trigger caspase-mediated apoptosis in human spermatozoa in vitro, suggesting a possible mechanism for CT-induced sperm damage [30]. Additional research points to a direct toxic effect of the bacteria on sperm cells, resulting in reduced functional capacity [12,13].

As for seminal volume, evidence from studies involving men with secondary infertility and sexually transmitted infections (STIs) has shown a trend toward lower semen volumes in STI-positive individuals, consistent with our observations in the context of CT infection [32]. This reduction may be attributed to ejaculatory duct obstruction or involvement of accessory glands, such as the seminal vesicles and prostate, as part of the infectious process [33].

The detection of CT can be carried out using several methods such as the detection of CT antigen by chromatography, antibodies by the ELISA method, and CT DNA by real-time PCR. Demonstration of antibodies may be delayed for several months; antibody titers may be low and many epithelial infections go undetected. There are many serological tests that are unable to recognize antibodies against various species of *Chlamydia* compared to PCR [34]. In our study, there was a single false-positive with ELISA assay. This is consistent with previous results that CT serological tests, though beneficial as a screening test, sometimes can lead to false positives due to cross reactivity with other bacterial antigens or non-specific binding [35]. Some patients with negative serological results were found positive by real-time PCR, indicating that real-time PCR offers superior sensitivity for the detection of active CT infection. Therefore, real-time PCR should be considered a more reliable diagnostic tool than ELISA for the identification of current infections, as serological assays primarily reflect past exposure rather than ongoing infection [36].

Persistent CT infection in the male genital tract is known to trigger a chronic inflammatory response, characterized by sustained production of pro-inflammatory cytokines. This immunological environment may contribute to epithelial damage and impaired sperm function [36]. CT has also evolved several immune evasion strategies [37]. It suppresses the production of chemotactic molecules and inflammatory cytokines, reducing immune cell recruitment. Furthermore, it interferes with antigen-presenting cells and downregulates the expression of MHC class I and II molecules, thereby limiting the activation of the adaptive immune response [38]. These mechanisms likely contribute to the persistence of infection and its long-term effects on male reproductive health.

In our study, we observed a statistically significant increase in the DNA fragmentation index (DFI) and Sperm Decondensation Index (SDI) in CT-positive patients compared to CT-negative controls (*p* < 0.01) (Figure 1). These findings are consistent with the results reported by Gallegos et al., who showed that men infected with CT and *Ureaplasma urealyticum* exhibited DFI and SDI levels 3.2 times higher than those of uninfected individuals [35]. In contrast, Sharma et al. reported increased DFI levels in infertile men with CT infection; however, their results did not reach statistical significance [39].

The increase in DNA damage observed in our study may reflect broader nuclear abnormalities in the spermatozoa of infertile men, including abnormal chromatin packaging, microdeletions, chromosomal rearrangements, aneuploidy, and DNA strand breaks [39]. Apoptotic pathways have been proposed as a major contributor to sperm DNA fragmentation, as highlighted by Sharma et al. [39,40]. As DNA integrity is closely associated with fertilization capacity and embryo development, our findings further support the hypothesis that CT infection may compromise male fertility through genomic damage.

Recent evidence also suggests a genetic predisposition to CT infection. Haratian et al. identified a single-nucleotide polymorphism (rs11467417) in the *DEFB126* gene—expressed primarily in the epididymal epithelium—which may increase susceptibility to CT infection and subsequent infertility in men [41]. Additionally, the severity and outcome of CT infection may depend on the bacterial serotype, host immune status, and microbiota composition. However, key questions remain, including the mechanisms of spontaneous clearance and the nature of host–pathogen interactions during persistent infection [42,43].

The current literature on the impact of CT on sperm quality is limited by small sample sizes, heterogeneous diagnostic methods (serology, culture, PCR, immunofluorescence), and differences in study populations. Our study addresses some of these limitations by analyzing a large cohort of 2731 semen samples and combining molecular (PCR) and serological techniques for CT detection, thereby enhancing the reliability and generalizability of our findings.

In conclusion, our results indicate that the presence of CT in semen is associated with increased sperm DNA fragmentation and chromatin decondensation, along with significant alterations in standard semen parameters, including volume, concentration, motility, morphology, and vitality. These findings underscore the importance of CT screening in the diagnostic evaluation of male infertility.

Our study has some limitations. Firstly, the analysis did not control for potential confounding conditions such as smoking, Body Mass Index (BMI), or the presence of co-existing urogenital infection, which may independently affect sperm quality and DNA fragmentation. Secondly, only a small number of samples were analyzed for DNA fragmentation due to resource constraints. These limitations must be considered while interpreting our results, and future research should attempt to include multivariable analyses and longitudinal follow-up.

## 5. Conclusions

This study underscores the relevance of CT infection as a potential contributor to male infertility through its association with compromised semen quality and sperm DNA integrity. These findings support the inclusion of molecular diagnostics, including PCR-based detection and sperm DNA fragmentation assessment, as part of the routine evaluation of male infertility, particularly in cases of unexplained infertility. Our findings underscore the need for routine molecular screening of CT in semen samples of infertile men, especially when standard semen analyses are inconclusive.

## Figures and Tables

**Figure 1 idr-17-00135-f001:**
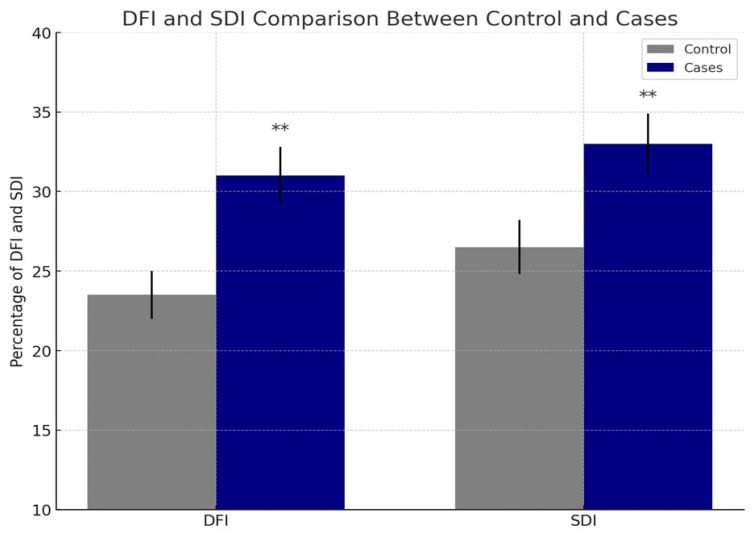
Variation in DNA fragmentation index (DFI) and chromatin decondensation index (SDI) between the case group and the control group. ** *p* < 0.01 (statistically highly significant). Cases: *n* = 273 patients with abnormal spermiogram, control: *n* = 59 patients with normal spermiogram.

**Table 1 idr-17-00135-t001:** Results of the statistical analysis of the spermiogram parameters in the different groups.

		Mean	Lower CI 95%	Higher CI 95%
Sperm Concentration (million/mL)	Control (678)	82.47	71.56	89.67
	CT+/Abnormal SG (273)	36 ^a^	29.18	38.64
	CT+/Normal SG (59)	74.28 ^b^	63.24	71.51
Total Motility Percentage (%)	Control (678)	71.38	66.27	77.59
	CT+/Abnormal SG (273)	38.54 ^a^	31.16	42.90
	CT+/Normal SG (59)	59.82 ^a,b^	54.46	61.07
Viability Percentage (%)	Control (678)	84.59	79.61	88.23
	CT+/Abnormal SG (273)	45.93 ^a^	37.18	51.71
	CT+/Normal SG (59)	73.65 ^a,b^	68.45	76.83
Percentage of Normal Morphology (%)	Control (678)	9.27	9.08	10.63
	CT+/Abnormal SG (273)	1.75 ^a^	1.18	2.92
	CT+/Normal SG (59)	8.42 ^a,b^	7.17	9.63
Leukocyte Concentration (Million/mL)	Control (678)	0.04	0.00	0.06
	CT+/Abnormal SG (273)	0.57 ^a^	0.46	0.72
	CT+/Normal SG (59)	0.09 ^b^	0.07	0.11

CI: Confidence interval. ^a^ Indicates significant difference between groups and control, *p* < 0.05. ^b^ A significant difference between patients with CT infection, abnormal SP, and normal SP, *p* < 0.05.

**Table 2 idr-17-00135-t002:** Prevalence of CT infection employing the detection of IgG antibodies and molecular methods.

	Abnormal SG	Normal SG	Total
ELISA	2.41% (57)	4.29% (102)	6.7% (159)
qPCR	3.25% (77)	4.05% (96)	7.3% (173)
Total	5.66% (134)	8.34% (198)	14% (332)

CT: Chlamydia trachomatis; ELISA: Enzyme-Linked Immunosorbent Assay; qPCR: real-time PCR.

## Data Availability

The data used to support the findings of this study are available from the corresponding author upon request.

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
