# Peer review of "Molecular Characterization of Chlamydia trachomatis Infection and Its Impact on Sperm Characteristics of Moroccan Infertile Men"

_2036-7449, 2025, doi:10.3390/idr17060135_

Round 1
Reviewer 1 Report
Comments and Suggestions for Authors
In the manuscript authors investigated the molecular detection of Chlamydia trachomatis (CT) and its association with sperm quality and DNA integrity among infertile Moroccan men. The use of both ELISA-based serological and qPCR-based molecular diagnostic methods strengthens the diagnostic accuracy. However, while the study includes a robust sample size and contributes region-specific data, several aspects of the manuscript require significant improvement to meet publication standards.
- Novel hypothesis of the work is not clear, it’s not clear how this work significantly advances the existing body of knowledge.
- Group classifications (control, normal SP, abnormal SP) are ambiguously defined, with unclear criteria and no baseline demographics provided.
- The use of a small subset of samples for DNA fragmentation analysis (e.g., only 59 controls out of 2572) lacks justification.
- The discussion is comprehensive but lacks a focused section on limitations. Important confounders such as smoking, BMI, or comorbid infections are not controlled or discussed.
- The authors mention a single false-positive by ELISA but did not discuss its implications and how often this occurs based on existing literature.
- The classification of groups (control, normal SP with CT+, abnormal SP with CT+) needs clearer description, especially how these were determined using semen parameters and diagnostic criteria.
Author Response
We would like to express our sincere gratitude to the reviewer for their thorough evaluation of our manuscript and for the insightful comments and suggestions provided. We highly appreciate the time and expertise devoted to the review process, which have contributed substantially to enhancing the clarity, scientific rigor, and overall quality of our work.
Comments 1 : Novel hypothesis of the work is not clear, it’s not clear how this work significantly advances the existing body of knowledge.
Response 1 : We thank the reviewer for this insightful comment. While we acknowledge that previous studies have investigated Chlamydia trachomatis infection among infertile men in Morocco, our work provides significant added value through the following aspects :
Our study hypothesizes that molecular detection of C. trachomatis enables a more accurate assessment of infection prevalence among infertile men and that this infection has a measurable impact not only on conventional semen parameters, but also on sperm DNA integrity. We further propose that even in men with normal standard semen analysis results, C. trachomatis infection can compromise DNA quality, which may explain idiopathic infertility cases.
Specific Contributions Compared to Previous Studies :
- Updated cohort and broader sampling :
Our study is based on a recent and relatively large cohort of Moroccan infertile men, providing updated prevalence estimates and clinical correlations relevant to current reproductive health trends. - Comprehensive evaluation of sperm quality :
Unlike earlier studies that focused mostly on basic semen parameters or prevalence, we performed a detailed statistical correlation between infection status and both conventional sperm parameters (motility, morphology, count, vitality) and sperm DNA fragmentation (DFI/SDI). This comprehensive assessment allows us to capture functional and molecular impairments in sperm quality. - Novel insight into DNA damage in asymptomatic patients :
we included the assessment of sperm DNA integrity (via DFI and SDI) in our study to explore potential subclinical effects of C. trachomatis infection. This approach may provide additional insight into cases of unexplained male infertility and reflects a growing consensus that advanced sperm function tests should complement conventional semen analysis, especially in the context of asymptomatic infections. - Clinical and diagnostic implications :
Our findings underscore the potential value of integrating both molecular pathogen screening and sperm DNA integrity testing into the diagnostic algorithm for male infertility. This is especially relevant in contexts like Morocco, where asymptomatic infections may go unnoticed and DNA integrity testing is not routinely performed. - Public health and contextual relevance :
By focusing on Moroccan men, our data contribute to a better understanding of regional patterns of male infertility and can inform national guidelines for STI screening and fertility management.
We have revised the Introduction section to make this novel positioning and our contribution to the field more explicit.
Comments 2 : Group classifications (control, normal SP, abnormal SP) are ambiguously defined, with unclear criteria and no baseline demographics provided.
Response 2 : Thank you for this important comment. We have clarified the group classifications by specifying the exact criteria for control, normal SP, and abnormal SP groups in the Methods section. Additionally, baseline demographic data for each group have been added to enhance transparency and interpretation
Comments 3 : The use of a small subset of samples for DNA fragmentation analysis (e.g., only 59 controls out of 2572) lacks justification.
Response 3 : The limited number of samples analyzed for DNA fragmentation was due to practical constraints such as assay cost, time, and resource availability. This analysis was intended as an exploratory sub-study to provide preliminary insights into the molecular impact of Chlamydia trachomatis infection on sperm DNA integrity. Despite the smaller sample size, the selected subset was representative of key groups and allowed meaningful comparisons. We acknowledge this limitation and suggest larger studies in the future to confirm these findings.
Comments 4 : The discussion is comprehensive but lacks a focused section on limitations. Important confounders such as smoking, BMI, or comorbid infections are not controlled or discussed
Response 4 : We thank the reviewer for this important observation. We acknowledge that the discussion did not include a clearly defined section on study limitations. In response, we have added a dedicated paragraph addressing key limitations, including the absence of control for potential confounders such as smoking status, body mass index (BMI), and other comorbid infections. These factors may influence semen parameters and sperm DNA integrity and should be considered in our future studies. We have also clarified that our results should be interpreted within the context of these limitations.
Comments 5 : The authors mention a single false-positive by ELISA but did not discuss its implications and how often this occurs based on existing literature.
Response 5 : We have added a brief discussion on this point in the revised manuscript to acknowledge the possibility of false positives and emphasize the importance of combining serological testing with molecular diagnostics for accurate detection.
Comments 6 : The classification of groups (control, normal SP with CT+, abnormal SP with CT+) needs clearer description, especially how these were determined using semen parameters and diagnostic criteria.
Response 6 : Thank you for your valuable comment. We have revised the Methods section to provide a clearer and more detailed description of the group classifications.

Reviewer 2 Report
Comments and Suggestions for Authors
General comments
This manuscript presents an important and clinically relevant investigation into the molecular detection of Chlamydia trachomatis (CT) and its impact on sperm parameters and DNA integrity in infertile Moroccan men. The use of both serological (ELISA) and molecular (qPCR) diagnostics, alongside sperm functional analyses (motility, morphology, vitality) and chromatin quality (DFI and SDI), provides a comprehensive overview of CT-associated male infertility. However, several aspects of the manuscript require significant improvement to enhance clarity, scientific rigor, and methodological transparency. The narrative often suffers from redundancy and lack of fluid transitions. Additionally, some methodological steps are not sufficiently justified (e.g., group classification, qPCR validation), and the discussion would benefit from a clearer structure and more cautious interpretation of associations. With revisions, this work can offer solid insights into the contribution of asymptomatic CT infections to idiopathic male infertility and the value of molecular screening in reproductive diagnostics.
Some specific comments:
The abstract is mostly well-structured, but the phrasing could be refined for conciseness and clarity. For instance: “...the presence of anti-CT antibodies by the Anti-CT IgG ELISA, confirmation of the presence of this bacteria...” for “...anti-CT IgG antibodies by ELISA, followed by qPCR confirmation of C. trachomatis DNA...”; “causes an alteration” for “is associated with significant alterations” to avoid causal overstatement.
The background is comprehensive, but too dense in some parts. Several concepts are introduced abruptly (e.g., leukospermia, oxidative stress, CT pathogenesis) without adequate transitions. I suggest consolidating paragraphs that discuss overlapping mechanisms (e.g., ROS production, DNA fragmentation, sperm apoptosis). The rationale for focusing on male infertility, despite the known female predominance in CT literature, is well-justified. However, the final paragraph would benefit from an explicit statement of the study hypothesis or expected outcomes.
It is not clear to me how the “normal SP infected” group was confirmed to be infection-positive without abnormalities — were these men entirely asymptomatic and normozoospermic?
Please clarify the number of replicates and how borderline/ambiguous results (e.g., single ELISA+/qPCR– cases) were treated. In Table 2, the discrepancy between ELISA and qPCR prevalence could be discussed more robustly — especially the high rate of ELISA–/qPCR+ cases, suggesting the utility of nucleic acid amplification for screening.
The discussion synthesizes literature effectively but is somewhat repetitive. Consider restructuring into thematic blocks (e.g., diagnostic approach; impact on sperm quality; DNA fragmentation; immune evasion by C. trachomatis; limitations). The statement that “our results are in perfect agreement with…” should be softened; science rarely supports “perfect” agreement. Several mechanistic claims (e.g., LPS-induced apoptosis, BTB disruption) are plausible but not directly tested here. Suggest rephrasing as “may contribute to...” or “have been proposed to explain...”.
Conclusion is appropriate but is too general. I suggest rewording to highlight:
“Our findings underscore the need for routine molecular screening of C. trachomatis in semen samples of infertile men, especially when standard semen analyses are inconclusive.”
Tables: Suggest including sample size (n) in footnotes.
Comments on the Quality of English Language
Minor revisions are needed. But overall the quality is adequate.
Author Response
We would like to express our sincere gratitude to the reviewer for their thorough evaluation of our manuscript and for the insightful comments and suggestions provided. We highly appreciate the time and expertise devoted to the review process, which have contributed substantially to enhancing the clarity, scientific rigor, and overall quality of our work.
Comments 1 : The abstract is mostly well-structured, but the phrasing could be refined for conciseness and clarity. For instance : “...the presence of anti-CT antibodies by the Anti-CT IgG ELISA, confirmation of the presence of this bacteria...” for “...anti-CT IgG antibodies by ELISA, followed by qPCR confirmation of C. trachomatis DNA...”; “causes an alteration” for “is associated with significant alterations” to avoid causal overstatement.
Response 1 : Thank you for your valuable suggestions. We have refined the abstract for improved clarity and conciseness by revising the phrasing as recommended.
Comments 2 : The background is comprehensive, but too dense in some parts. Several concepts are introduced abruptly (e.g., leukospermia, oxidative stress, CT pathogenesis) without adequate transitions. I suggest consolidating paragraphs that discuss overlapping mechanisms (e.g., ROS production, DNA fragmentation, sperm apoptosis). The rationale for focusing on male infertility, despite the known female predominance in CT literature, is well-justified. However, the final paragraph would benefit from an explicit statement of the study hypothesis or expected outcomes.
Response 2 : We have revised the background section to improve clarity by consolidating overlapping content related to ROS production, DNA fragmentation, and sperm apoptosis, and by adding smoother transitions between concepts. We have included an explicit statement of the study hypothesis and expected outcomes at the end of the Introduction, as recommended.
Comments 3 : It is not clear to me how the “normal SP infected” group was confirmed to be infection-positive without abnormalities — were these men entirely asymptomatic and normozoospermic ?
Response 3 : The “normal SP infected” group consisted of men who tested positive for Chlamydia trachomatis by qPCR but showed no clinical symptoms typically associated with infection (e.g., urethral discharge, dysuria), and had semen parameters within normal reference ranges according to WHO 2021 guidelines. These men were asymptomatic and identified during routine fertility assessments, highlighting the presence of subclinical infections.
Comments 4 : Please clarify the number of replicates and how borderline/ambiguous results (e.g., single ELISA+/qPCR– cases) were treated. In Table 2, the discrepancy between ELISA and qPCR prevalence could be discussed more robustly — especially the high rate of ELISA–/qPCR+ cases, suggesting the utility of nucleic acid amplification for screening.
Response 4 : Thank you for this valuable observation. We have clarified in the revised manuscript that all PCR analyses were performed in duplicate to ensure result reliability. In cases of discrepancy or borderline results (e.g., ELISA-positive/qPCR-negative), we considered qPCR as the more specific method due to its higher sensitivity and specificity for detecting Chlamydia trachomatis DNA. The single ELISA+/qPCR– case was interpreted as a potential false-positive serological result, likely due to antibody persistence or cross-reactivity.
We have also expanded the discussion around Table 2 to highlight the high frequency of ELISA–/qPCR+ cases. This finding supports the added value of molecular testing in detecting asymptomatic infections that may not trigger a detectable immune response.
Comments 5 : The discussion synthesizes literature effectively but is somewhat repetitive. Consider restructuring into thematic blocks (e.g., diagnostic approach; impact on sperm quality; DNA fragmentation; immune evasion by C. trachomatis; limitations). The statement that “our results are in perfect agreement with…” should be softened; science rarely supports “perfect” agreement. Several mechanistic claims (e.g., LPS-induced apoptosis, BTB disruption) are plausible but not directly tested here. Suggest rephrasing as “may contribute to...” or “have been proposed to explain...”.
Response 5 : Thank you for your thoughtful and constructive feedback. We agree that restructuring the discussion into thematic blocks will improve clarity and reduce redundancy. We have revised the section accordingly, organizing it into the following themes: diagnostic approach; impact on sperm quality; DNA fragmentation; immune evasion by C. trachomatis; and limitations of the current study.
Comments 6 : Conclusion is appropriate but is too general. I suggest rewording to highlight :
“Our findings underscore the need for routine molecular screening of C. trachomatis in semen samples of infertile men, especially when standard semen analyses are inconclusive.”
Response 6 : We have revised the conclusion to emphasize the importance of routine molecular screening for Chlamydia trachomatis in semen samples,
Comments 7 : Tables : Suggest including sample size (n) in footnotes.
Response 7 : Thank you for the suggestion. We have added the sample size (n) in the footnotes of all relevant tables for greater clarity.

Reviewer 3 Report
Comments and Suggestions for Authors
This research focuses on the effects of Chlamydia trachomatis on the semen quality. Authors used different methodological approaches to analyse the effects of bacterial infection on semen quality, but sometimes the results obtained are not clearly described. Therefore, deep revision is necessary before the acceptance of the manuscript for publication. Please, note that all sections need deep improvement to adjust to the requirements of a scientific document.
Moreover, the manuscript contains several grammatical and typographical errors that need deep revision and improvement.
Specific comments that authors should also address are detailed below:
-Please, refer to “viability” instead of "vitality” on the text.
-Line 105. Please, indicate the sample size of the SP group.
-Line 220. “Noted” is not an appropriated term in this content. Author should refer to as “observed”.
-Lines 220-222. Please, revise and improve the content of these lines and give a clear message.
-In line 222 authors differentiate between normal and abnormal SP groups. Nevertheless, these two groups are not properly described in the Material and Methods section. Authors should include in the Material and Methods a clear description of the three groups of patients included in the study, by indicating the sample size and age range, was well as the criteria used to classify the patients into each group.
-In Table 1, authors provide the sample size of the control group, but they do not of abnormal and normal SP groups. Please provide give the sample size of these two groups.
-Still in Table 1, the authors refer to “Percentage of morphology”, but it is unclear what do the authors refer to. I suppose that it refers to the percentage of morphologically normal spermatozoa; if so, please change the use an appropriate term for this parameter and provide an accurate description in the Material and Methods section.
-Similarly, the sperm parameter “Motility percentage” is quite confusing since it is not clear if authors refer to either the percentage of total motility spermatozoa or the percentage of progressive motility spermatozoa. This should be clearly indicated in the Table 1 and in the Material and Methods section.
-Please, add in the Material and Methods section a brief description of how the sperm viability was measured.
-Lines 233-241. Please, revise and improve the content of these lines and provide a clear message.
-Figure 1 should be improved. I strongly recommend the authors to eliminate the green background. In the Figure legend, please indicate the meaning of the two asterisks, and the sample size of Control and Cases. On the other hand, what does “Cases” mean? Which group of patients does it correspond to? Please, use an appropriate term.
-Lines 266-267. Please, refer to sperm concentration, sperm motility, sperm viability and sperm morphology.
-Line 269. Please, provide the percentage of asymptomatic patients.
-Line 270. Please, correct as “In symptomatic patients, it leads to subacute urethritis”.
-Line 274. Please, refer to sperm concentration, sperm motility, sperm viability and sperm morphology.
-Line 276. Please, eliminate the words “perfect” and “team”.
-Lines 275-279. Please, revise and improve the content of these and give a clear message.
-The information given in the Discussion section is quite interesting, but the whole section needs deep revision and improvement to fulfil the standards of a scientific report.
-Line 317. Please, eliminate the words “perfect” and “team”.
-Please, revise the Conclusions section and avoid repeating again the results obtained.
Author Response
We would like to express our sincere gratitude to the reviewer for their thorough evaluation of our manuscript and for the insightful comments and suggestions provided. We highly appreciate the time and expertise devoted to the review process, which have contributed substantially to enhancing the clarity, scientific rigor, and overall quality of our work.
Comments 1 : Please, refer to “viability” instead of "vitality” on the text.
Response 1 : We have replaced all instances of "vitality" with "viability" throughout the manuscript
Comments 2 : Line 105. Please, indicate the sample size of the SP group
Response 2 : Thank you for your observation. We have now indicated the sample size of the SP group in line 105 for clarity.
Comments 3 : Line 220. “Noted” is not an appropriated term in this content. Author should refer to as “observed”.
Response 3 : We have replaced the term “noted” with “observed” in line 220 to improve clarity and appropriateness of the language.
Comments 4 : Lines 220-222. Please, revise and improve the content of these lines and give a clear message.
Response 4 : Thank you for your helpful suggestion. We have revised lines 220–222 to improve clarity and ensure the message is conveyed more precisely.
Comments 5 : In line 222 authors differentiate between normal and abnormal SP groups. Nevertheless, these two groups are not properly described in the Material and Methods section. Authors should include in the Material and Methods a clear description of the three groups of patients included in the study, by indicating the sample size and age range, was well as the criteria used to classify the patients into each group.
Response 5 : Thank you for pointing this out. We agree that a clear description of the patient groups is essential. Accordingly, we have revised the Materials and Methods section to include a detailed description of the three groups analyzed in the study. This includes the sample size, age range, and the specific criteria used to classify patients into each group (normal sperm parameters, abnormal sperm parameters, and control).
Comments 6 : In Table 1, authors provide the sample size of the control group, but they do not of abnormal and normal SP groups. Please provide give the sample size of these two groups
Response 6 : Thank you for your observation. We have updated Table 1 to include the sample sizes of both the normal and abnormal sperm parameter (SP) groups, in addition to the control group.
Comments 7 : Still in Table 1, the authors refer to “Percentage of morphology”, but it is unclear what do the authors refer to. I suppose that it refers to the percentage of morphologically normal spermatozoa; if so, please change the use an appropriate term for this parameter and provide an accurate description in the Material and Methods section.
Response 7 : The term in Table 1 indeed refers to the percentage of morphologically normal spermatozoa, according to WHO criteria. We have corrected the terminology in Table 1 to “Normal morphology (%)” for clarity, and we have added a clear description in the Materials and Methods section indicating that this value reflects the percentage of spermatozoa with normal morphology
Comments 8 : Similarly, the sperm parameter “Motility percentage” is quite confusing since it is not clear if authors refer to either the percentage of total motility spermatozoa or the percentage of progressive motility spermatozoa. This should be clearly indicated in the Table 1 and in the Material and Methods section.
Response 9 : The term in Table 1 indeed refers to the percentage of total mobility according to WHO criteria. We have corrected the terminology in Table 1 to “Total mobility (%)” for clarity, and we have added a clear description in the Materials and Methods section indicating that this value reflects the percentage of spermatozoa with total mobility
Comments 10 : Please, add in the Material and Methods section a brief description of how the sperm viability was measured.
Response 10 : A brief description of the sperm viability assessment method using eosin 2% staining has been added to the Materials and Methods
Comments 11 : Lines 233-241. Please, revise and improve the content of these lines and provide a clear message.
Response 11 : Thank you for the comment. The paragraph (lines 233–241) has been revised for clarity and to better highlight the differences between ELISA and qPCR results.
Comments 12 : Figure 1 should be improved. I strongly recommend the authors to eliminate the green background. In the Figure legend, please indicate the meaning of the two asterisks, and the sample size of Control and Cases. On the other hand, what does “Cases” mean? Which group of patients does it correspond to? Please, use an appropriate term.
Response 12 : Thank you for your detailed and constructive feedback regarding Figure 1. In response:
- We have removed the green background from the figure to improve visual clarity and eliminate unnecessary visual elements.
- The figure legend has been revised to indicate the meaning of the two asterisks (**), which represent statistically significant differences (p < 0.01).
- We have also added the sample sizes of both the Control group (patients with normal spermogram, n = 59 patients) and the Cases group (patients with abnormal spermogram, n = 273) in the legend.
Comments 13 : Lines 266-267. Please, refer to sperm concentration, sperm motility, sperm viability and sperm morphology.
Response 13 : We have revised the sentence to explicitly refer to all evaluated sperm parameters. The text now clearly mentions sperm concentration, sperm motility, sperm viability, and sperm morphology, in accordance with your suggestion.
Comments 14 : Line 269. Please, provide the percentage of asymptomatic patients.
Response 14 : Thank you for your valuable comment. We have now included the percentage of asymptomatic patients in the revised manuscript.
Comments 15 : Line 270. Please, correct as “In symptomatic patients, it leads to subacute urethritis”.
Response 15 : The sentence has been corrected to: In symptomatic patients, Chlamydia trachomatis infection leads to subacute urethritis…..to improve clarity and accuracy.
Comments 16 : Line 274. Please, refer to sperm concentration, sperm motility, sperm viability and sperm morphology.
Response 16 : We have revised the sentence to explicitly refer to all evaluated sperm parameters. The text now clearly mentions sperm concentration, sperm motility, sperm viability, and sperm morphology, in accordance with your suggestion.
Comments 17 : Line 276. Please, eliminate the words “perfect” and “team”.
Response 17 : We eliminated them from the manuscript
Comments 18 : Lines 275-279. Please, revise and improve the content of these and give a clear message.
Response 18 : Thank you for the suggestion. The sentence has been revised for improved clarity and scientific tone.
Comments 19 : The information given in the Discussion section is quite interesting, but the whole section needs deep revision and improvement to fulfil the standards of a scientific report
Response 19 : Thank you for your comment. The Discussion section has been thoroughly revised to improve its structure, clarity, and scientific rigor. We have reorganized the content, reduced repetition, and ensured that interpretations are well aligned with our findings.
Comments 20 : Line 317. Please, eliminate the words “perfect” and “team”.
Response 20 : We eliminated them from the manuscript
Comments 21 : Please, revise the Conclusions section and avoid repeating again the results obtained
Response 21 : We thank the reviewer for this helpful suggestion. The Conclusions section has been revised to avoid repetition of the results

Reviewer 4 Report
Comments and Suggestions for Authors
An interesting study by Kabbour et al., based on a large patient cohort.
However, this manuscript requires several major revisions before it can be considered for publication.
The organization of study groups needs to be clarified, as the current structure lacks consistency and makes the study difficult to follow. In the Methods section, the authors indicate that patients are initially divided into a control group (normal sperm count) and a case group (abnormal sperm count). Then, patients in the case group who are CT-positive become the “abnormal SP” group. However, in the Results section, the reported prevalence of CT infection (between 2.41% and 3.25% based on ELISA and qPCR) does not align with the definition of the SP-abnormal group as CT-positive.
To improve clarity and interpretability, I recommend reclassifying the population into two independent criteria:
-
Spermogram status: Normal / Abnormal
-
CT status: Negative / Positive
This would lead to four distinct and interpretable subgroups:
-
Normal SG / CT-negative
-
Normal SG / CT-positive
-
Abnormal SG / CT-negative
-
Abnormal SG / CT-positive
Such a structure would make your results more understandable and allow for clearer statistical comparisons.
The data in Table 2 are unexpected: CT infections appear more frequent in patients with normal sperm parameters, and test performance (ELISA vs qPCR) seems inconsistent between groups. Could the authors provide an explanation for this finding?
Additionally, if you are evaluating the performance of qPCR, it cannot simultaneously serve as the gold standard. Please clarify the reference method used for diagnostic comparison.
It would also be useful to provide national prevalence data on Chlamydia trachomatis infections in Morocco (for both men and women) to contextualize your findings.
I also have minor comments :
Introduction:
-
Please rephrase the first sentence on Page 2, Line 44.
-
Write Chlamydia trachomatis in full at first mention, then use “CT” throughout the rest of the manuscript.
-
The acronym “STI” is defined twice (Page 2, Lines 58 and 62). Please remove one.
-
The introduction contains an excessive number of paragraph breaks. Please harmonize the formatting for smoother reading.
Methods:
-
In section 2.3 (Molecular tests), there is no need to explain the basic principles of PCR. Focus instead on the specific procedures, primers, and validation methods used in your study.
Results:
-
Page 7, Line 224: No need to restate the definition of the groups in this sentence
- Page 7, Line 249: DFI et SDI are already defined in the Methods section
Discussion:
-
The first paragraph reiterates too many result details. Focus on interpretation rather than repetition.
-
Page 7, Line 304: Based on your current data, the conclusion drawn here appears unsupported. Please revise.
- Page 8, Line 318: Specify which Mycoplasma species is being referred to. Is it Mycoplasma genitalium?
Tables and Figures :
Table 1:
-
Consider adding a dedicated column for p-values. Including statistical significance notes only in the table legend is not sufficient.
Table 2:
-
Please spell out ELISA as Enzyme-Linked Immunosorbent Assay.
-
Define the “q” in qPCR (e.g., quantitative PCR or real-time PCR) and ensure consistency throughout the manuscript.
Figure 1:
-
It would be clearer and more logical to present DFI and SDI values for control and case groups side by side.
-
Clarify what the asterisks above the right-side bars represent (e.g., statistical significance?).
-
Remove the green background—it adds visual noise without conveying information.
Author Response
We would like to express our sincere gratitude to the reviewer for their thorough evaluation of our manuscript and for the insightful comments and suggestions provided. We highly appreciate the time and expertise devoted to the review process, which have contributed substantially to enhancing the clarity, scientific rigor, and overall quality of our work.
Comments 1 : To improve clarity and interpretability, I recommend reclassifying the population into two independent criteria :
- Spermogram status : Normal / Abnormal
- CT status : Negative / Positive
This would lead to four distinct and interpretable subgroups:
- Normal SG / CT-negative
- Normal SG / CT-positive
- Abnormal SG / CT-negative
- Abnormal SG / CT-positive
Such a structure would make your results more understandable and allow for clearer statistical comparisons.
Response 1 : We thank the reviewer for this insightful suggestion. Following your recommendation, we have reclassified the study population according to two independent criteria—spermogram status (normal vs. abnormal) and Chlamydia trachomatis infection status (positive vs. negative)—resulting in four clearly defined subgroups. This approach enhances the clarity and interpretability of our analyses.
The abnormal spermogram/CT-negative group was identified but not included in the detailed analysis, as our primary focus was to investigate the impact of Chlamydia trachomatis infection on sperm quality. Excluding this group allowed us to specifically assess the differences attributable to CT infection among men with either normal or abnormal spermograms.
Comments 2 : The data in Table 2 are unexpected: CT infections appear more frequent in patients with normal sperm parameters, and test performance (ELISA vs qPCR) seems inconsistent between groups. Could the authors provide an explanation for this finding?
Response 2 : Asymptomatic carriage and early infection : CT is known for its ability to persist in the male genital tract without causing overt symptoms or immediate alterations in sperm quality. Some of the individuals with normal sperm profiles but positive qPCR results may be in the early or latent phase of infection, before measurable sperm damage occurs.
Sensitivity difference between techniques : ELISA detects host antibodies (IgM/IgG), which reflect past or ongoing immune responses, while qPCR identifies bacterial DNA directly.
Study population characteristics : The inclusion of a large number of subfertile men from couples with unexplained infertility could lead to a higher-than-expected detection of CT in normozoospermic individuals, revealing infections that may affect fertility through mechanisms not captured by standard sperm analysis (e.g., DNA damage or immune-mediated effects).
Comments 3 : Additionally, if you are evaluating the performance of qPCR, it cannot simultaneously serve as the gold standard. Please clarify the reference method used for diagnostic comparison.
Response 3 : We fully agree that using qPCR both as the method under evaluation and as the reference standard would introduce bias. In our study, qPCR was not used to evaluate its own performance but rather served as the reference method for diagnostic comparison, given its widely recognized higher sensitivity and specificity in the detection of Chlamydia trachomatis, as supported by numerous studies The performance of ELISA was assessed against qPCR, which we considered the most appropriate reference standard for current infection due to its direct detection of bacterial DNA
Comments 4 : It would also be useful to provide national prevalence data on Chlamydia trachomatis infections in Morocco (for both men and women) to contextualize your findings.
Response 4 : We have now included national prevalence data on Chlamydia trachomatis (CT) infections in Morocco to provide better epidemiological context for our findings.
Introduction
- Comments 1 : Please rephrase the first sentence on Page 2, Line 44.
Response 1 : We have revised the sentence for clarity and scientific accuracy.
- Comments 2 : Write Chlamydia trachomatisin full at first mention, then use “CT” throughout the rest of the manuscript.
Response 2 : We have now written "Chlamydia trachomatis" in full at its first mention and used the abbreviation "CT" consistently throughout the remainder of the manuscript.
- Comments 3 : The acronym “STI” is defined twice (Page 2, Lines 58 and 62). Please remove one.
Response 3 : We have removed the redundant definition of the acronym “STI” to improve clarity.
- Comments 4 : The introduction contains an excessive number of paragraph breaks. Please harmonize the formatting for smoother reading.
Response 4 : The introduction has been reformatted to reduce the number of paragraph breaks, resulting in improved flow and readability.
Methods :
Comments 1 : In section 2.3 (Molecular tests), there is no need to explain the basic principles of PCR. Focus instead on the specific procedures, primers, and validation methods used in your study.
Response 1 : We have revised Section 2.3 to remove the general explanation of PCR principles
Results :
- Comments 1 : Page 7, Line 224: No need to restate the definition of the groups in this sentence
Response 1 : We have removed the group classifications in the indicated sections to improve clarity and avoid repetition.
- Comments 2 : Page 7, Line 249: DFI et SDI are already defined in the Methods section
Response 2 : We have removed the redundant definitions of DFI, SDI
Discussion :
Comments 1 : The first paragraph reiterates too many result details. Focus on interpretation rather than repetition.
Response 1 : Thank you for your valuable suggestion to focus more on interpretation and reduce the repetition of detailed results in the first paragraph. We understand the importance of clarity and avoiding redundancy. However, based on feedback from another reviewers emphasizing the inclusion of key data for completeness, we have made efforts to strike a balance by condensing the detailed results while retaining the most critical information to support the interpretations. This revision aims to ensure the paragraph remains concise yet informative, facilitating readers’ understanding without unnecessary repetition. We hope this approach meets your expectations.
Comments 2 : Page 7, Line 304 : Based on your current data, the conclusion drawn here appears unsupported. Please revise.
Response 2 : We thank the reviewer for this valuable comment. In response, we have revised the manuscript to better reflect the findings and current evidence. Our data indicate that patients with negative serological tests were positive by qPCR, supporting the conclusion that PCR has superior sensitivity and specificity for detecting active Chlamydia trachomatis infection compared to ELISA. Accordingly, the text now emphasizes PCR as a more reliable diagnostic method for active infection rather than merely a complementary technique. The manuscript has been updated to incorporate these clarifications
Comments 3 : Specify which Mycoplasma species is being referred to. Is it Mycoplasma genitalium ?
Response 3 : Thank you for your helpful comment. We have clarified the text to specify the species involved. In this study, the species referred to Ureaplasma urealyticum. The manuscript has been updated accordingly to avoid ambiguity.
Figures and tables :
Table 1 :
- Consider adding a dedicated column for p-values. Including statistical significance notes only in the table legend is not sufficient
Response : Thank you for your suggestion. However, we believe that adding a dedicated column for p-values may lead to confusion in this case, as two different statistical comparisons are being reported: (a) between each group and the control, and (b) between patient subgroups with abnormal and normal semen parameters. To avoid misinterpretation, we have opted to indicate statistical significance using superscript letters within the table and explain them clearly in the legend. We believe this approach ensures clarity while maintaining a concise table format.
Table 2:
- Please spell out ELISA as Enzyme-Linked Immunosorbent Assay.
- Define the “q” in qPCR (e.g., quantitativePCR or real-time PCR) and ensure consistency throughout the manuscript.
Response : We have spelled out ELISA as "Enzyme-Linked Immunosorbent Assay" at its first mention and defined “qPCR” as “quantitative PCR” consistently throughout the manuscript to ensure clarity.
Figure 1:
- It would be clearer and more logical to present DFI and SDI values for control and case groups side by side.
- Clarify what the asterisks above the right-side bars represent (e.g., statistical significance?).
- Remove the green background—it adds visual noise without conveying information.
Response : Thank you for your valuable suggestions. In response, we have modified the graph to present DFI and SDI values side by side for the control and case groups, in order to enhance clarity and allow for a more direct comparison between the groups.
We have clarified in the figure legend that the asterisks above the right-side bars indicate statistical significance (p < 0.05 or p < 0.01, as appropriate). Additionally, we have removed the green background from the figure to reduce visual distraction and improve overall readability.

Round 2
Reviewer 1 Report
Comments and Suggestions for Authors
Thank you to the authors for their thoughtful and detailed responses to the reviewer comments. The revised manuscript is much improved in terms of clarity, scientific quality, and overall presentation.
The authors now clearly present their central hypothesis—that Chlamydia trachomatis infection may compromise sperm DNA integrity even in men with normal semen parameters, potentially contributing to unexplained infertility. This idea is supported by a solid dataset and the use of both conventional semen analysis and molecular tools (qPCR, DFI, SDI), which together strengthen the study’s conclusions.
The Methods section has been clarified, especially in terms of group classification and diagnostic criteria. I also appreciate the new section on study limitations, which acknowledges important confounding factors like BMI, smoking, and co-infections. The brief discussion on ELISA limitations and the justification for the subset used in DNA fragmentation analysis are also helpful additions.
Author Response
We sincerely thank the reviewer for these encouraging and constructive comments. We are pleased to know that the revised manuscript has been found to be improved in terms of clarity, scientific rigor, and overall presentation.
We also appreciate your recognition of the revised structure, particularly the clearer presentation of our central hypothesis, the refined Methods section, and the newly added discussion on study limitations. Your thoughtful feedback has been instrumental in strengthening the manuscript, and we are grateful for the time and attention you devoted to its review.
Reviewer 3 Report
Comments and Suggestions for Authors
Authors introduced nearly all the modifications requested and so the overall quality of the manuscript has been improved. Nevertheless, the document still contains several typographical errors that should be corrected during the editing process.
Other minor changes that authors should address are detailed below:
Lines 33 and 34. Please, refer to “spermiogram” instead of "spermogram”. Authors should revise the whole document and correct this term.
Line 62. Authors should add the reference number instead of the surname of the first author.
Line 165. Please, refer to “following alterations” instead of "following”.
Line 147. Please, refer tot “total motility” instead of “total mobility”. Please, revise the whole document and correct this term.
Line 152. Please, refer to “Sperm viability” instead of "Viability”.
Line 152. Please, refer to "(viable)” instead of "(live).
Line 237. Please, refer to “ranging from 24 to 56 years old”.
Lines 285 and 286. Please, refer to "years old” instead of "years”.
Lines 309-319. Please, revise the content of these lines and avoid repeating “one study”. Please, note that this is not a proper to compare the results between different studies.
Line 327. Please, refer to “previous results” instead of “reported previously”.
Author Response
Thank you for your positive feedback. We carefully revised the manuscript to correct the remaining typographical errors through both manual proofreading and spell-checking.
Lines 33–34 – Please, refer to “spermiogram” instead of "spermogram”. Authors should revise the whole document and correct this term.
Response:
Corrected as requested. The term “spermogram” has been replaced with “spermiogram” throughout the manuscript.
Line 62 – Authors should add the reference number instead of the surname of the first author.
Response:
The author's surname has been removed and replaced with the appropriate reference number.
Line 147 – Please, refer to “total motility” instead of “total mobility”. Please, revise the whole document and correct this term.
Response:
The term “total mobility” has been corrected to “total motility” in the relevant sections of the manuscript.
Line 152 – Please, refer to “Sperm viability” instead of "Viability”.
Response:
Revised accordingly. “Viability” now appears as “Sperm viability”.
Line 152 – Please, refer to "(viable)” instead of "(live)”.
Response:
Corrected as requested: “(live)” has been replaced with “(viable)”.
Line 165 – Please, refer to “following alterations” instead of "following”.
Response:
Modified. The phrase now reads “following alterations”.
Line 237 – Please, refer to “ranging from 24 to 56 years old”.
Response:
Updated as suggested. The sentence now indicates “ranging from 24 to 56 years old”.
Lines 285–286 – Please, refer to "years old” instead of "years”.
Response:
Done. “Years” has been replaced with “years old” as appropriate.
Lines 309–319 – Please, revise the content of these lines and avoid repeating “one study”. Please, note that this is not a proper to compare the results between different studies.
Response:
This section has been entirely revised to improve scientific tone and avoid repetitive phrasing. The revised text synthesizes findings from multiple studies without direct comparison.
Line 327 – Please, refer to “previous results” instead of “reported previously”.
Response:
Amended. “Reported previously” has been replaced by “previous results”.
Reviewer 4 Report
Comments and Suggestions for Authors
The revisions made by the authors significantly enhance the quality of the manuscript.
I do not believe it is necessary for the authors to include the size and age of the groups in the Methods section; this information would be more appropriately placed at the beginning of the Results section.
Additionally, the term Chlamydia trachomatis appears again on page 3, lines 128–129. Please use the abbreviation “CT” instead.
Finally, in the Introduction (page 2, line 62), the authors should cite reference numbers only, without including the names of the first authors.
Author Response
Thank you for your positive feedback.
Comment 1:
I do not believe it is necessary for the authors to include the size and age of the groups in the Methods section; this information would be more appropriately placed at the beginning of the Results section.
Response:
We thank the reviewer for this helpful remark. In the current version, we chose to include the size and age of the study groups in the Methods section in response to previous reviewers feedback, which recommended clearly stating the demographic characteristics of each group before presenting the results. This structure was adopted to enhance clarity and allow the reader to interpret the findings in the appropriate context.
That said, if you consider it necessary, we would be glad to move this information to the beginning of the Results section.
Comment 2:
Additionally, the term Chlamydia trachomatis appears again on page 3, lines 128–129. Please use the abbreviation “CT” instead.
Response:
Corrected as requested. We have replaced “Chlamydia trachomatis” with the abbreviation “CT” at the indicated location.
Comment 3:
In the Introduction (page 2, line 62), the authors should cite reference numbers only, without including the names of the first authors.
Response:
Amended accordingly. The surname of the first author has been removed, and only the reference number is now cited in this sentence.